# EFFICIENT ON-DEVICE AGENTS VIA ADAPTIVE CONTEXT MANAGEMENT

## ABSTRACT

On-device AI agents offer the potential for personalized, low-latency assistance, but their deployment is fundamentally constrained by limited memory capacity, which restricts usable context. This reduced practical context window creates a trade-off between supporting rich, stateful interactions with complex tool capabilities and maintaining on-device feasibility. We break this trade-off with a framework for context-efficient on-device agents, driven by three synergistic optimizations (1) a dynamic memory system using specialized LoRA adapters to distill conversational history into a compressed, and structured *Context State Object*; (2) a minimalist serialization format for tool schemas to minimize token overhead per tool; and (3) a just-in-time schema-passing mechanism that loads full tool definitions only upon tool selection. We instantiate this framework by adapting a 3B parameter SLM to context-efficient trajectories and rigorously evaluate it against a conventional baseline on complex user tasks. Our agent matches, or exceeds, the performance of a conventional baseline while dramatically compressing context, achieving more than a 6-fold reduction in initial system prompt context and a 10- to 25-fold reduction in context growth rate based on the interaction verbosity, demonstrating that strategic context management is key to unlocking capable and persistent on-device AI.

## 1 INTRODUCTION

AI agents, powered by Large Language Models (LLMs), have catalyzed productivity gains across numerous domains (Wang et al., 2025b; Ju & Aral, 2025). Their true potential, however, lies in migrating from the cloud to personal devices, where they can leverage private data and control local applications to unlock a new frontier of personalized productivity (Ferev et al., 2024; Chen et al., 2024). This transformative vision, however, is fundamentally limited by the on-device hardware constraints, creating a significant bottleneck for progress (Xu et al., 2024). Central to this bottleneck is the agent's context window, which serves as its working memory and is essential for sophisticated capabilities like maintaining conversation state and utilizing complex tools. Maintaining a long context on-device creates two key challenges: **(1)** unsustainable memory consumption from the linearly growing ($\mathcal{O}(n)$) KV cache (Xiao et al., 2024), and **(2)** degraded reliability with longer inputs (Liu et al., 2023). This context bottleneck is amplified by three core aspects of agentic behavior. First, **conversational history** in the user–agent–environment loop accumulates rapidly through token-heavy tool calls and environment responses, obscuring essential information from earlier turns (Zhou et al., 2025b). Second, the **verbosity of tool schemas** (e.g., OpenAPI (OpenAPI Initiative, 2024) format) floods the context with thousands of tokens. Third, this creates a **tool scalability** problem, limiting the agent's versatility (Erdogan et al., 2024). The result is a crippled agent which is either forgetful and unable to maintain a coherent dialogue due to frequent context dumping, or simplistic and incapable of performing complex tasks. Existing solutions are ill-suited for this domain as they are designed for inter-session memory management (Packer et al., 2024), incompatible with optimizations like KV-caching (Zhou et al., 2025b), use heuristics for context management (Kim et al., 2025), or are designed for very long contexts of 100K+ tokens Lin et al. (2025). In contrast, on-device agents face tight memory budgets, where even 5–10K tokens can be prohibitive, yet preserving conversational detail and flow is essential.

In this work, we present a framework for building context-efficient on-device agents that targets each source of context bloat. We introduce a dual-adapter memory system that uses a specialized LoRA

adapter (Hu et al., 2021) to distill conversational history into a compressed, structured *Context State Object (CSO)*. To tackle tool-related overhead, we propose a two-pronged tool management system: a minimalist serialization format for schemas and a just-in-time mechanism that loads definitions only upon selection. The developed models achieve strong task performance at a fraction of the token cost. We instantiate our framework by fine-tuning a 3B parameter SLM and rigorously evaluate it against a conventional baseline on complex user tasks. **Our primary contributions are:**

- A novel on-device agent architecture for local tool execution or routing to cloud LLM.
- A dual-adapter memory system that maintains a compressed, but interpretable, conversational state which matches the performance of strong baselines on complex, multi-turn tasks while reducing the context growth rate by a factor of 10 to 25, with larger gains for verbose conversations.
- An integrated tool management system that combines token-efficient schemas with selective tool-passing resulting in more than 6x lower initial tool overhead for more capable agents.

## 2 RELATED WORKS

### 2.1 ON-DEVICE AGENTS AND FUNCTION CALLING

The drive for privacy, low latency, and personalization has spurred a shift towards on-device intelligence, built upon Small Language Models (SLMs) like Gemma-3 and Phi-3-mini that can operate within device constraints (Xu et al., 2024; Team et al., 2025; Abdin et al., 2024). The frontier for these SLMs lies in specialized agentic applications, where they act as low-cost orchestrators or directly interact with the environment via function calling (Chen et al., 2024; Lee et al., 2025). A common pattern is using an SLM as a router to dispatch tasks to other components (Yue et al., 2025; Tran et al., 2025). Our work adopts this hybrid on-device/cloud routing paradigm, but focuses on the context efficiency necessary for practical, long-running interactions.

Modern function calling agents, built on foundational frameworks like ReAct and Toolformer (Yao et al., 2023; Schick et al., 2023), are typically trained with Supervised Fine-Tuning (SFT) on large-scale synthetic datasets (Prabhakar et al., 2025; Qin et al., 2023). While large models achieve high accuracy, performance degrades for the SLMs used on-device (Zhang et al., 2025; Abdelaziz et al., 2024; Lin et al., 2024). A primary driver of this is the token overhead from tool schemas. Recent work confirms that reducing irrelevant tools improves performance (Paramanayakam et al., 2024), and systems like TinyAgent use retrieval to select a tool subset before prompting (Erdogan et al., 2024). Our work builds on this by introducing a two-stage process where the agent first selects a tool from a lightweight list, allowing it to be fully aware of its capabilities at minimal token overhead, and only then loads its full, token-optimized schema.

### 2.2 CONTEXT AND MEMORY MANAGEMENT

Context management has been studied through multiple lenses. Inter-session approaches such as recursive summarization (Wang et al., 2025a) and MemGPT (Packer et al., 2024) emphasize persistence but often lose fine-grained detail. Intra-session methods like MEM1 (Zhou et al., 2025b) improve fidelity but remain computationally intensive and incompatible with KV caching (Pope et al., 2022). Long-context modeling efforts (e.g., retrieval-augmented generation (Lin et al., 2025), sparse attention (Xu et al., 2025)) extend sequence length to 100K+ tokens, but target different scales than on-device budgets. Another direction compresses or removes KV cache entries (Kim et al., 2025; Park et al., 2025), but relies on low-level heuristics, such as attention magnitude and reconstruction error, which risk discarding key details that may only be needed much later in the trajectory (e.g., an address or ID). These compressed representations are opaque and hard to debug. In contrast, we condense history at the semantic level, where a trained model maintains a compact and rich log, ensuring interpretable, task-relevant memory that remains reliably accessible for tool use while achieving stronger context reductions.

## 3 FRAMEWORK FOR CONTEXT-EFFICIENT ON-DEVICE AGENTS

In this section, we detail the architecture and mechanisms of our proposed framework for building context-efficient on-device agents, describing the overall system architecture, then elaborate on our two core context-reduction contributions, and finally describe our data generation pipeline.

## 3.1 SYSTEM ARCHITECTURE

The foundation of our framework is an on-device agent powered by a 3B-parameter SLM. This agent acts as an orchestrator, as illustrated in Figure 1, and performs one of three actions based on user input: (1) **Direct Response** for general conversation; (2) **Local Execution** by calling on-device tools for tasks like checking a calendar; or (3) **Cloud Delegation** for complex queries.

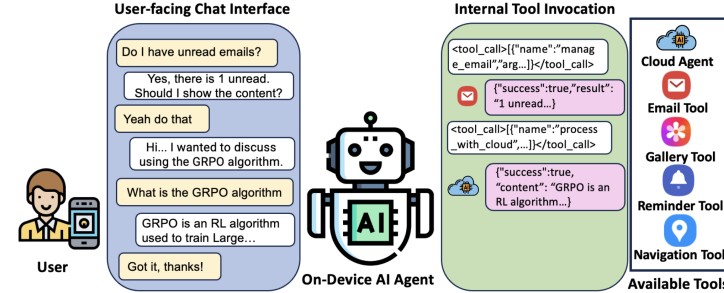

Figure 1: System architecture of the on-device AI agent. The user interacts with the agent through a chat interface. The on-device agent processes the user's requests and performs internal tool invocations, utilizing a suite of local, on-device tools (e.g., Email, Gallery, Reminders) as well as a more powerful Cloud Agent for complex queries.

This model leverages local speed and privacy while ensuring complex problems can be solved effectively with the Cloud Agent. The Cloud Agent, which is a much larger and capable LLM, also prevents common user frustrations, such as incorrectly calling irrelevant tools, hallucinations or denying requests due to lack of capability. However, the verbose cloud responses intensify the need for extreme context efficiency to maintain high performance and low memory usage on-device.

## 3.2 DYNAMIC MEMORY VIA CONTEXT STATE OBJECT

To manage conversational history without high token costs, we introduce a dynamic memory system built on a dual-adapter architecture. This system replaces the raw conversation history with a compressed, structured text log we term the *Context State Object (CSO)*. The CSO serves as an evolving, append-only log that captures the essential state of the interaction in a token-efficient format. This mechanism is powered by two complementary, parameter-efficient LoRA (Hu et al., 2021) modules:

- **The Executor ($\text{LoRA}_{\text{Exec}}$):** The primary agentic adapter, fine-tuned to perform the core tasks in Section 3.1 by leveraging the CSO for conversational context.

- **The State-Tracker ($\text{LoRA}_{\text{Mem}}$):** A specialized, lightweight (119M parameters) adapter which generates a concise, append-only CSO update based on the latest turn.

The memory update cycle occurs after each assistant turn, designed for maximum efficiency:

1. **Task Execution:** The `Executor` adapter is active and processes the user query and the state $CSO_{t-1}$, to generate a response or tool call.

2. **State Update Generation:** The active adapter is swapped to the `State-Tracker`. It is prompted with $CSO_{t-1}$ and the latest user-assistant exchange to generate only the new lines to be appended (the delta, $\Delta_t$).

3. **State Concatenation:** The new state object, $CSO_t$, is formed by appending the delta to the previous state: $CSO_t = CSO_{t-1} \oplus \Delta_t$.

This architecture, as demonstrated in Figure 2a provides several advantages for stateful, on-device agents. First, by generating a small, append-only update once per turn, it maintains a linear growth of context but with a significantly reduced rate. Our experiments show this leads to a 10-fold reduction in the rate of context increase for typical, non-cloud conversations, and a 25-fold reduction with verbose outputs from the cloud-based agent. Over conversations involving 20+ assistant turns, this translates into savings of thousands of tokens, enabling more capable and persistent interactions.

Second, the design is optimized for inference. The lightweight `Executor` and `State-Tracker` adapters can be simultaneously loaded into memory with minimal overhead. Crucially, our custom inference implementation tracks the Key-Value (KV) caches for both adapters separately. Because the CSO is only appended to, the KV cache for the large, static portion of the prompt (system

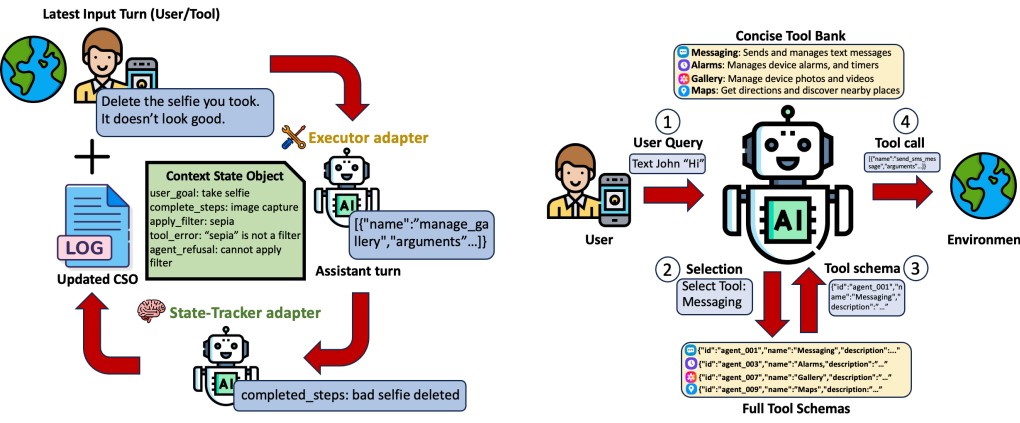

(a) Stateful context management using CSO.    (b) Efficient tool selection from a concise tool bank.

Figure 2: Our two context optimization techniques for on-device agents. (a) A state-tracking system maintains a compact history. (b) A two-step tool-call process reduces the overhead of tool schemas.

instructions, tool schemas (for the Executor), and $CSO_{t-1}$) is preserved for both adapters. Only the small update and new query require processing, dramatically accelerating inference speed.

Finally, the State-Tracker is explicitly trained to produce a rich, semi-structured log that selectively preserves key task-relevant information, including detail extracted from otherwise verbose cloud responses, thereby maintaining fidelity at substantially lower token cost. We use a key-value format as we hypothesize it will leverage the base model's strong aptitude for structured data, as developed through function-calling training. This format avoids the need for complete sentences and through explicit training allows the State-Tracker to log critical details often lost in conventional summarization, such as tool errors, agent refusals, and other environmental feedback. This continuous, detailed log allows the Executor, which is co-adapted to this format, to better track implied user intent, key requirements and recover from errors, leading to a more robust and natural conversational flow. A walkthrough is provided in Section A.8. This system incurs a modest overhead of an approximate 80MB memory increase and a 500ms latency per turn for the CSO update cycle (Galaxy S25 CPU). We believe optimized deployments can further decrease these costs.

### 3.3 TOOL SCHEMA MANAGEMENT

To address the context bloat due to tool schemas, we utilize a two-pronged strategy that tackles both the verbosity of individual tool representations and the scaling of tools.

**Token-Efficient Schema Representation**    Standard API formats such as the OpenAPI Specification (OpenAPI Initiative, 2024) are notoriously verbose for LLM prompts, in part due to human-oriented formatting and whitespace. Prior work and tooling often employ schema *minification* to reduce token cost, typically by stripping indentation or unused fields. Building on this, we train our models directly with a compact format that retains only the essential fields for function invocation (name, description, parameters) while removing non-essential whitespace. As shown in Table 4 (Appendix), this yields an average **40%** reduction in tokens per tool. Crucially, this optimization is integrated into our broader framework for context and state management, allowing significantly more tools to fit within a given context window and thereby enhancing the agent's usable capabilities.

**Just-in-Time Schema Passing**    While our efficient format reduces the cost of each tool, including all schemas in the prompt remains unscalable and introduces significant prompt bloat with irrelevant context (Paramanayakam et al., 2024). Conversely, a pure retrieval-based approach that injects multiple relevant tool schemas at each turn (Erdogan et al., 2024) presents its own trade-offs wherein it can paradoxically consume context faster than standard methods, and can cause the agent to fail when the retriever returns no tools or irrelevant ones. To address these trade-offs, we implement a hybrid *just-in-time (JIT)* schema passing mechanism (Figure 2b). Our two-step approach ensures the agent is always aware of its full capabilities at a minimal token cost, while only loading relevant full schemas:

1. **Selection Stage:** The system prompt provides the agent with a lightweight list of all available tool names and their concise descriptions. This makes the agent fully aware of its potential actions. Based on the user's query, its task is to select the most appropriate tool by outputting its name (e.g., Timer tool) along with the reasoning for selection.

2. **Execution Stage:** Upon selection, the full, token-optimized tool schema is provided to the agent in the subsequent observation. Now, the agent's task is to generate the correct arguments in the tool call based on the user's query and the provided schema.

This JIT approach provides a highly scalable solution to tool management. The initial context remains small and manageable, approximately **25%** of the baseline size, allowing the agent to be equipped with a diverse set of capabilities without overwhelming the on-device SLM. This drastic reduction in initial prompt size also leads to approximately **5x** reductions in Time-To-First-Token (TTFT) for on-device deployment[1]. Together, our efficient schema format and JIT mechanism form a complete system that dramatically reduces tool-related context overhead.

### 3.4 DATA GENERATION AND TRAINING PROTOCOL

To train our agent to operate effectively within our proposed framework, we develop a comprehensive pipeline to produce a high-quality dataset and fine-tune the agent's dual-adapter memory system and two-stage tool use, along with the baseline which follows standard fine-tuning practices.

**On-Device Use Case Generation** Our process adapts the APIGen-MT pipeline (Prabhakar et al., 2025) for the on-device domain. To promote generalizability and prevent overfitting, we first create a diverse pool of 100 base on-device tools. For each base tool, we then use an LLM to generate 50 variants with modifications to parameters and descriptions, resulting in 5,000 unique tools. We defined four task categories for generation: multi-tool, cloud delegation, mixed on-device/cloud, and conversational queries. We use Gemini 2.0 Flash (Google Cloud, 2024) for multi-turn trajectory generation. For each scenario, a detailed rubric provides the model with a sampled toolset, an initial user utterance, a persona, and a ground-truth tool sequence (more details in Appendix A.7). Separate instances of Gemini then simulate the entire interaction, playing the roles of a non-deterministic user, the assistant, and a tool execution environment that returns realistic success or failure messages. The generated trajectories undergo a rigorous two-stage curation process. They are first validated against the ground-truth tool sequence for task completion. Successful trajectories are then evaluated by an LLM-as-a-judge (also Gemini 2.0 Flash) to filter out low-quality or unnatural interactions.

**Fine-tuning and Specialized Adapter Training** We use LoRA (Hu et al., 2021) to efficiently fine-tune the 3B xLAM 2 base model (Zhang et al., 2025), adapting it to our on-device framework while preserving its strong foundational capabilities. Additionally, the LoRA approach is also modular, allowing our specialized adapters to be composed with other potential modules, such as safety guardrails or further personalization layers. For training, each example provides a context of 10-14 tools (the correct tool, Cloud tool, and distractor tools) to teach robust selection and delegation. All training is conducted using our token-efficient tool schema format. We also qualitatively test the performance on-device with 4-bit quantized models. For training the State-Tracker adapter, the teacher model is used to generate CSO updates for every assistant turn.

## 4 EXPERIMENTAL SETUP

To rigorously evaluate our context-efficient framework, we design a comprehensive evaluation to assess both generalization and performance across a diverse range of on-device scenarios.

### 4.1 EVALUATION DATASET AND BASELINES

We design a challenging evaluation set of 406 on-device tasks using the generation pipeline from Section 3.4, but with a crucial modification where the test set is built using 19 new, unseen tools for

---

[1]Evaluated on CPU-based Samsung Galaxy S25 devices and averaged across 3 evaluations. The reduction in TTFT is an expected consequence of the drastically smaller initial prompt size, as prefill processing time is a primary driver of initial latency.

testing generalizability (examples provided in Appendix A.4). This ensures we measure the agent's ability to apply learned skills to novel APIs, not just memorize tool signatures. We use a simulated user to interact with the trained models over long trajectories, introducing ambiguity, multi-step intents, and naturalistic expression to make the evaluation realistic (Vijayvargiya et al., 2025; Zhou et al., 2024; 2025a). We evaluate five model variants to isolate the impact of our contributions:

- **xLAM 2 3B (Reference):** The model from previous work (Zhang et al., 2025) prompted with full conversational history and OpenAPI tool schemas.
- **Baseline (FT):** The baseline model fine-tuned on our use case, but only with token-efficient schemas and full conversational history.
- **Tool-Efficient:** Fine-tuned with our complete tool schema management system.
- **Memory-Efficient:** Fine-tuned with our dual-adapter memory system for history management.
- **Combined:** Our full model, integrating both the tool and memory efficiency mechanisms.

## 4.2 Evaluation Metrics

We evaluate model performance using a suite of automated and qualitative metrics.

- **Rule-Based Tool Call Metrics:** To provide a more nuanced view of the agent's tool-use capabilities, we measure the quality of its tool calls.
  - *Precision* measures the fraction of tool calls made by the agent that were correct (i.e., part of the ground-truth sequence). It penalizes non-existent, syntactically incorrect, or irrelevant calls.
  - *Recall* measures the fraction of required ground-truth tool calls that the agent successfully made.
  - *F1 Score* is the harmonic mean of Precision and Recall, providing a single, balanced measure of tool-calling accuracy.
- **LLM-as-Judge:** We evaluate the qualitative nature of the trajectories to identify if the user's goal was easily satisfied with minimal frustration on a scale of 1-5 with 5 being an ideal conversation.

We also track the input context length at each assistant turn across the evaluated models. To mitigate variability from the non-deterministic user simulation, performance on all metrics is averaged across three runs. Results are broken down by task category for a holistic analysis.

## 5 Results

In this section, we present the empirical evaluation of our framework, first analyzing task performance to demonstrate that our efficiency mechanisms preserve or enhance capability, and then quantifying the dramatic reductions in token overhead.

## 5.1 Task Performance

We evaluate our fine-tuned models against the xLAM-2 3B model to contextualize our results. Table 1 presents a detailed performance breakdown across our complex task categories. While the xLAM-2 model excels at generic tool calling, our fine-tuning helps adapt to the nuanced task of cloud orchestration, as shown by the Baseline's superior performance on cloud-related tasks. Among our proposed methods, the Tool-Efficient model demonstrates the highest precision, a direct result of its deliberate two-stage process, though this rigidity impacts its recall. The Memory-Efficient model, however, emerges as a robust performer. It achieves a perfect F1 score on Cloud Delegation and the highest F1 on the challenging On-Device + Cloud tasks, while also maintaining strong multi-tool performance. Its high qualitative (Q) scores in these categories further suggest a more coherent and effective user interaction. Finally, our Combined model achieves the highest overall tool-calling F1 scores, showing strong decision making and intelligence. Its qualitative performance in tool-calling trajectories lags behind the strongest variants because the multi-step selection and invocation turns lead to a higher ratio of structured tool call turns in the training data relative to user-facing information presentation turns. This imbalance causes the model to prioritize structured tool calls over natural responses, impacting interaction quality. We keep the data sampling method fixed across models to ensure a fair comparison without introducing confounding factors.

Table 1: Performance breakdown on complex, multi-turn tasks. We compare our fine-tuned models and also present the xLAM-2 3B model as a reference. We report F1-Score (F1), Precision (P), Recall (R), and a qualitative score (Q, 1-5 scale). Higher scores are better across all metrics.

| Model Variant | Cloud-Tool Only | | | | Multi-Tool | | | | On-Device + Cloud | | | | Convo. |
|---|---|---|---|---|---|---|---|---|---|---|---|---|---|
| | F1 | P | R | Q | F1 | P | R | Q | F1 | P | R | Q | Q |
| xLAM-2 3B | 0.41 | 0.53 | 0.63 | 2.07 | 0.87 | 0.86 | 0.91 | 3.59 | 0.57 | 0.65 | 0.63 | 2.27 | 2.18 |
| Baseline (FT) | **1.00** | **1.00** | **1.00** | 3.79 | 0.83 | 0.79 | 0.92 | 3.17 | 0.88 | 0.94 | 0.87 | 3.45 | 2.87 |
| Tool-Efficient | 0.93 | 0.97 | 0.95 | 3.55 | 0.86 | **0.92** | 0.88 | **3.61** | 0.73 | 0.85 | 0.74 | 3.32 | 3.21 |
| Mem-Efficient | **1.00** | **1.00** | **1.00** | 4.11 | 0.87 | 0.87 | 0.90 | 3.37 | 0.89 | **0.95** | 0.88 | **3.64** | 3.32 |
| **Combined** | 0.99 | 0.98 | **1.00** | 2.68 | **0.93** | 0.92 | **0.95** | 3.32 | **0.94** | 0.92 | **0.92** | 3.00 | **3.80** |

## 5.2 Analysis of Context Efficiency

Our framework's primary motivation is to drastically reduce context overhead. While we have shown our methods perform similarly, or better than standard finetuning, here we examine the initial prompt size and the subsequent rate of context growth to highlight the token cost savings.

### 5.2.1 Initial Context Size and Tool Scalability

The initial prompt size is critical for on-device performance, directly impacting Time-To-First-Token (TTFT) and determining the maximum number of tools an agent can be aware of. As shown in the starting point (Assistant Turn 0) of Figure 3, the models have vastly different initial footprints.

- The **xLAM-2** model, using standard schemas, starts with an initial context of 3,200 tokens.
- Our **Baseline** and **Memory-Efficient** models, which use our token-efficient schema format, reduce this initial cost to 2,100 tokens.
- Our **Tool-Efficient** model, which includes only tool names and descriptions in the initial prompt and instead relies on JIT loading of tool schemas, requires a mere 400 tokens.
- The **Combined** model leverages the strengths of the Tool-Efficient model in initial context.

This demonstrates that our JIT mechanism and schema format, allow an agent to be aware of an approximately 8x larger toolset for the same token budget and significantly improves TTFT.

### 5.2.2 Context Growth Rate over Conversation

We now analyze context accumulation over a conversation. Figure 3 shows that for general Multi-Tool use, the xLAM-2 and Baseline models' contexts grow linearly to 5,000 tokens. The Tool-Efficient model, while starting lowest, also sees its context expand to comparable sizes on long trajectories. In stark contrast, the Memory-Efficient model's context remains remarkably flat, increasing by only 100-200 tokens after 25 turns, leading to a 10-fold reduction in context accumulation compared to the Baseline. The Combined model sees slightly higher growth of around 500-700 tokens due to full schemas being injected into its inputs unlike the Memory-Efficient model.

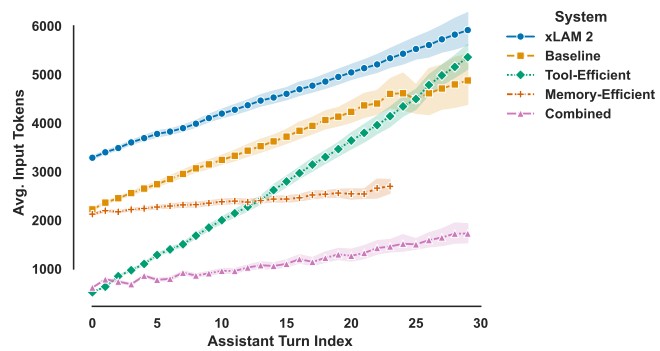

Figure 3: Averaged context input length over assistant turns for the **Multi-Tool** category in the evaluation runs. Shaded regions represent 95% Confidence Intervals.

This advantage becomes even more extreme in Cloud Delegation scenarios (Figure 4). The Baseline's context explodes to 10,000-12,000 tokens due to verbose cloud responses whereas

the Memory-Efficient model distills these responses into concise CSO updates, limiting its context growth to 500 tokens. We achieve a context growth that is more than 20-fold smaller.

The Combined model leverages the strengths of the Memory-Efficient model to maintain low context growth rates but at much lower absolute token counts. The slower context growth of the xLAM-2 and Tool-Efficient models in the longest cloud trajectories is a byproduct of their failure modes. The former often fails to call the cloud tool correctly, while the latter occasionally gets stuck in select-loops (unlike the Baseline which gets stuck in cloud tool call loops), thus avoiding its verbose response from cloud. Therefore, our Memory-Efficient model is unique in that it achieves its extreme context efficiency while achieving the highest success rate.

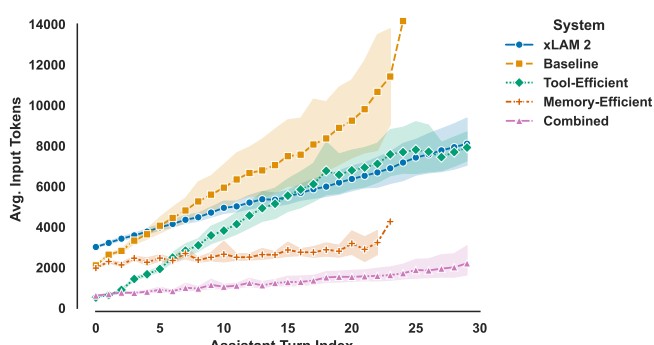

Figure 4: Averaged context input length over assistant turns for the **Cloud Delegation** category in the evaluation runs. Shaded regions represent 95% Confidence Intervals.

## 6 DISCUSSION

Our empirical results demonstrate that our context-efficient framework dramatically reduces token overhead while maintaining or enhancing agent performance on complex tasks. The strong results of our Combined model, in particular, highlight our central hypothesis: for on-device agents, strategic context management is a critical enabler of stateful behavior. In this section, we interpret these findings and analyze the architectural trade-offs revealed by our experiments, focusing on the Tool-Efficient and Memory-Efficient models as the Combined model directly borrows traits from both.

### 6.1 SHORTER CONTEXTS BOOST DECISION-MAKING AND COHERENCE

While our quantitative results show a clear performance gap on complex tasks, a qualitative analysis reveals the underlying reasons. The Context State Object (CSO) enables our Memory-Efficient agent to exhibit more robust long-term decision-making. Unlike a simple summary, the CSO functions as an explicit, structured memory, allowing the agent to effectively track previous actions and observations, and recover from errors due to less distractions for attention in the input.

A common failure mode for other model variants is repetitive error looping. When a tool call fails, the agent often ignores the error from the tool and attempts the same failing action on its next turn, especially when the preceding conversational history is long. We hypothesize this is a symptom of attention dilution (Liu et al., 2023), where in a long, unstructured context, the critical error feedback from the previous turn can be drowned out by less relevant historical tokens. In contrast, the State-Tracker adapter is trained to distill critical events, such as tool_error, into a concise, structured entry in the CSO. When prompted with this updated state, the Executor can reliably attend to the logged error, and pivot to a new strategy, such as retrying, clarifying with the user or delegating to the cloud. This is reflected in the performance metrics, where the recall of Tool-Efficient model is boosted in the Combined model due to better error recovery with the CSO.

### 6.2 EFFICACY OF DISTILLED STATE TRACKING

A key finding is the effectiveness of the State-Tracker adapter, trained solely via supervised fine-tuning. A qualitative analysis of its generated Context State Objects (CSOs) reveals that it learned to perform more than simple summarization. We observe three notable properties:

1. **Task-Oriented State Decomposition:** The State-Tracker learns to decompose ambiguous user utterances into a structured state representation. For instance, for a conditional request like *Run*

*a scan if battery more than 50%*, it generates distinct entries for the *user_goal* and the blocker (*need_info*). Crucially, it also logs the Executor's limitations, such as adding *agent_limitation: cannot check battery level* after a failure. This creates a detailed and concise log of events and the keys capture the higher level flow, enabling the agent to avoid repeating erroneous actions.

2. **Adaptive Logging:** The model learns to vary the length of its CSO updates based on the information density of the preceding turn. It generates concise updates for simple turns but more detailed updates after verbose cloud responses, suggesting a capability to gauge information salience.

3. **Alignment with Base Abilities:** The key-value format of the CSO has proved highly effective, likely because it aligns with the base xLAM-2 model's strong training on structured data. We hypothesize the keys (e.g., *user_goal*) function as strong attentional anchors, making it easier for the Executor to understand the general flow of the conversation compared to unstructured prose.

While this process is inherently lossy, due to discarding the conversational filler, our results show it preserves the task-critical information and conversational flow for on-device deployment.

### 6.3 PRECISION VS. RECALL TRADE-OFF

Our study reveals an interesting architectural trade-off between the Tool-Efficient and Memory-Efficient models. The Tool-Efficient model, with its two-stage selection process, consistently achieving high precision as compared to its recall. Its rigid structure, which requires explicit reasoning to justify tool selection, appears to enforce a more deliberate initial choice but comes at the cost of lower recall, as the model struggles to pivot to an alternative tool if its first choice is incorrect. Further, we observe that the complexity of the two-step protocol can sometimes lead to degraded instruction following, making the model less responsive to subtler user requirements.

The Memory-Efficient model, conversely, demonstrates greater flexibility and higher overall recall. Its access to a rich conversational history via the CSO allows it to better handle ambiguity and attempt different strategies if one approach fails, especially as exact previous turns are not visible preventing repetitive looping Xu et al. (2022). A precision-focused JIT approach may be preferable in high-stakes scenarios where a single incorrect tool call has persistent negative consequences, whereas high recall provides the crucial flexibility needed to ensure a user's ultimate goal is achieved in more complex or exploratory interactions. The Combined model benefits from the interaction of the two strategies as it achieves high precision due to the 2-step tool calling, and high recall due to the ability to attempt different strategies if initially unsuccessful.

## 7 CONCLUSION, LIMITATIONS, AND FUTURE WORK

By systematically addressing context bloat from both conversational history and tool schemas, we introduce a holistic framework that enables agents to be both more efficient and more capable. Our Combined agent, with a dual-adapter memory system and integrated tool management protocol, supports complex, stateful interactions at a fraction of the token cost of conventional methods. Experiments show strong performance on challenging multi-turn tasks while reducing context growth by over an order of magnitude, making practical on-device deployment far more feasible.

At the same time, our study has natural boundaries. To clearly isolate the effects of our context mechanisms, training was limited to a single epoch on a fixed data distribution, leaving open how broader training regimes or diverse domains might impact results. Our method also builds on supervised distillation from a teacher model, which constrains the State-Tracker's quality to the teacher's capabilities. Finally, because existing benchmarks are not designed for dynamic context mechanisms, we present our results on a custom evaluation protocol. While this limits direct comparability, we ground our findings by benchmarking against a strong function-calling baseline.

These limitations also suggest promising directions for future work. Reinforcement learning could enable agents to make more adaptive decisions, overcoming tool-calling and clarification patterns learned during supervised fine-tuning. Additionally, evaluating our context management approaches across open-domain tasks would test their generalizability. Overall, our results underscore that achieving persistent, capable on-device agents will rely on continued advances in computational efficiency.

## REPRODUCIBILITY STATEMENT

To ensure the reproducibility of the presented results, this paper provides comprehensive details on the methodology, data generation, and experimental setup. All key components of the proposed framework are described with the intention of enabling replication by an independent research group.

- **Framework Architecture:** The designs of the dual-adapter memory system and the just-in-time tool management protocol are detailed in Section 3. A step-by-step walkthrough of the custom Key-Value cache management logic is provided in Appendix A.8.
- **Data Generation and Curation:** The full prompts used to guide the teacher model (Gemini 2.0 Flash) for generating both the conversational scenarios and the Context State Object updates are included in Appendix A.7 and A.3. The two-stage data curation process is described in Section 3.4.
- **Training and Evaluation Protocol:** All hyperparameters for LoRA and model fine-tuning are listed in the tables within Appendix A.2, including details on the base model, learning rates, and training data composition. The evaluation protocol, including the composition of the custom test set with unseen tools and the definition of all metrics, is described in Section 4. To account for variability in the non-deterministic evaluation, all reported performance metrics are the average of three independent runs.

The comprehensive details provided throughout the main paper and appendices are intended to be sufficient for the full re-implementation and validation of our work.

## LLM USAGE

We used a large language model to assist with polishing the writing style, condensing the content, and improving clarity. All research ideas, methods, experiments, and analyses were developed and conducted by the authors. The LLM did not contribute to scientific content.

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

## A  APPENDIX

### A.1  EXAMPLE ON-DEVICE TOOLS

To ground our methodology in concrete examples, this section provides a representative sample of the on-device tools used in our evaluation dataset. These examples illustrate the diversity of the agent's capabilities, from simple device controls to more complex application interactions and the crucial cloud delegation tool. Each tool is defined by a unique ID, a descriptive name, and a structured function schema.

---

**Example: Messaging Tool**

**Description:** Sends and manages text messages or chats via installed messaging apps.
**Function:** manage_messages(action, recipient, content, app_preference (optional), ...)

---

**Example: Gallery & Photo Tool**

**Description:** Manages photos and videos in the device gallery: search, view, share, or perform quick edits.
**Function:** manage_gallery_items(action, search_keywords (optional), item_identifier (optional), ...)

---

**Example: Navigation Tool**

**Description:** Provides directions and finds nearby points of interest based on locally stored map data.
**Function:** get_local_navigation_info(request_type, destination_or_poi_type, ...)

---

**Example: Ride Service Launcher**

**Description:** Launches a ride service app (Uber or Lyft) with the destination pre-filled.
**Function:** launch_ride_service_app(destination, origin_address, ...)

---

**Example: Cloud Delegation Tool**

**Description:** ESCALATION: For complex queries, creative tasks, deep reasoning, or requests requiring extensive external knowledge that on-device agents cannot handle.
**Function:** process_with_cloud_llm(user_query_context, reason_for_escalation)

---

### A.2  TRAINING SPECIFICATIONS

This section provides a detailed overview of the hyperparameters and settings used to train our models, ensuring full reproducibility of our results. All models were trained using the LlamaFactory framework (Zheng et al., 2024) on a single NVIDIA A6000 GPU with 48GB of VRAM.

### A.2.1 BASE MODEL AND QUANTIZATION

The base model for all our experiments is a 3B-parameter Small Language Model (SLM), xLAM 2 (Zhang et al., 2025), specifically chosen for its strong foundational capabilities in function calling. For final deployment on-device, all fine-tuned models, including the baseline, were quantized to 4-bit precision using llama.cpp library Gerganov & llama.cpp contributors (2023) to simulate a realistic on-device resource footprint.

### A.2.2 LoRA HYPERPARAMETERS

We used Low-Rank Adaptation (LoRA) for all parameter-efficient fine-tuning. The same hyper-parameters were applied consistently across all LoRA adapters (Executor, State-Tracker, and the adapters for the Baseline and Tool-Efficient models) to ensure a fair comparison. The key LoRA settings are detailed in Table 2. The total number of trainable parameters for each LoRA adapter was approximately 119M.

Table 2: Key LoRA hyperparameters used for training all adapters.

| Hyperparameter | Value |
|---|---|
| LoRA Rank ('r') | 64 |
| LoRA Alpha ('alpha') | 128 |
| LoRA Dropout | 0.05 |
| Target Modules | All Linear Layers ('all') |

### A.2.3 FINE-TUNING HYPERPARAMETERS

All models were fine-tuned for a single epoch over the training dataset. The training dataset consists of approximately 150,000-200,000 samples for the different tasks. We used the AdamW optimizer and a cosine learning rate scheduler. The primary fine-tuning hyperparameters are listed in Table 3.

Table 3: Primary hyperparameters used for the fine-tuning process.

| Hyperparameter | Value |
|---|---|
| Epochs | 1 |
| Learning Rate | 1e-4 |
| Optimizer | AdamW |
| Learning Rate Scheduler | Cosine |
| Warmup Ratio | 0.1 |
| Batch Size | 32 (with gradient accumulation) |
| Maximum Sequence Length | 4096 |

### A.2.4 TRAINING DATA COMPOSITION

Our training dataset consists of five distinct categories, designed to teach the agent a diverse range of skills. The sampling ratio for each category during training was carefully chosen to emphasize more complex, multi-step interactions, as detailed below:

- **Multi-Tool Tasks:** 40%
- **On-Device then Cloud Tasks:** 20%
- **Conversational Tasks:** 10%
- **Single-Tool Tasks:** 15%
- **Cloud-Only Complex Tasks:** 15%

This distribution ensures that the model receives ample training on the challenging, stateful scenarios where our context-efficiency mechanisms provide the most benefit. All training is conducted using our token-efficient tool schema format.

### A.2.5 EVALUATION DATA COMPOSITION

The intents for evaluation are distributed across the five task categories to provide a holistic performance picture:

- **Multi-Tool Tasks:** 181 intents requiring a sequence of multiple on-device tool calls.
- **Cloud Delegation Tasks:** 75 intents designed to be out-of-scope for on-device tools, requiring correct delegation to the cloud agent.
- **Mixed On-Device and Cloud Tasks:** 100 intents that involve a combination of local tool use and cloud delegation.
- **Conversational Tasks:** 50 intents where the correct behavior is to converse naturally without calling any tools.

### A.2.6 EVALUATION DATASET

For the purpose of this work, we are not able to evaluate our developed models on existing function calling benchmarks like ToolBench (Qin et al., 2023), Berkeley Function-Calling Benchmark (Patil et al., 2025), etc. as we modify the the trajectory building to support more context efficient formats and our models will not work with the provided context of these benchmarks. Thus, we have an open-ended evaluation with a user with a goal and the models can adapt the context as per their own requirements to accomplish the goal.

### A.3 PROMPT FOR STATE-TRACKER (LoRA$_{\text{MEM}}$) GENERATION

The following is the complete prompt structure provided to Gemini 2.0 Flash to generate the ground-truth Context State Object (CSO) updates. This process, referred to as hard distillation, forms the training data for our 'State-Tracker' adapter. The prompt is designed as a zero-shot request, guided by strict rules and a few-shot demonstration of the desired append-only behavior to ensure high-quality, concise state updates.

**State-Tracker Generation Prompt**

You are an expert AI assistant that analyzes new conversation turns and generates a concise, append-only log entry. Your output MUST ONLY be the new lines to be appended to a checklist. DO NOT repeat any information already present in the "Full Checklist So Far". You are responsible for maintaining a summary of the conversation so far such that all important details are captured concisely. We do not want to degrade performance, but we want to be token efficient.

**Your Rules:**

- Be Extremely Concise: Use single words for keys (e.g., 'user_goal').
- Log Core Facts: If new, add the user's 'user_goal' or any 'agent_refusals'.
- Log Progress: If a step is completed, add it under 'completed_steps' but do not make it repetitive (adding 'completed_step' for every task done).
- Log persistent, unresolved tool errors concisely to prevent agent from making the same mistake again.
- You can format the new lines to append as you want (with or without keys) with the overall goal that the new lines capture the most information in the most concise manner.
- Our primary goal is creating a very concise summary but preserving all the key details of the conversation so no other messages are required for reference.
- You can create some new fields (do NOT overdo it) in the checklist such that crucial information to the task is kept track of.
- You DO NOT need to add the summarized user or tool message, just longer horizon information about the task that will help in later turns.
- DO NOT output the full checklist. Only output the new lines.
- If there are no new lines you wish to append, output this text without anything else:
  `# NO_UPDATE`
- Follow these examples perfectly:

---

**Example 1: Generating the first checklist entries**
Full Checklist So Far:
# This is the start of the conversation.
New Messages:
user: Predict the stock market prices for Google tomorrow.
assistant: I can't provide specific stock market predictions. However, I can search for information related to Google's stock and market trends.
New Lines to Append:
user_goal: get analyst predictions for google stock
agent_refusals: cannot predict stock prices

**Example 2: Appending a completed step** Full Checklist So Far:
user_goal: enable power saving mode
New Messages:
assistant: "name":"manage_network_settings","arguments":"action":"toggle_wifi"
tool: "status":"success","action":"toggle_wifi","new_state":"disabled"
New Lines to Append:
completed_steps: wifi disabled

---

**Now, perform the task for the following input:**
`[Full Checklist So Far]`
*{The full text of the current CSO is injected here}*
`[New Messages]`
*{The text of the new user, tool, and assistant turns is injected here}*
`[New Lines to Append]`

## A.4 Example Evaluation Scenarios

This section details representative examples from the primary task categories used in our test set. Each example includes the high-level goal, the initial prompt given by the simulated user, and the scenario notes which guided the trajectory generation.

---

**Example Evaluation Intents by Category**

CATEGORY: MULTI-TOOL TASKS

*These scenarios require the agent to execute a sequence of multiple on-device tools.*

---

**Goal:** Play a specific song and then adjust the volume.
**Initial Utterance:** "Play 'Bohemian Rhapsody' and turn it up."
**Scenario Notes:** Tests audio control. Requires playing a song and then adjusting the volume. AI needs to understand that 'turn it up' means increasing the volume.

---

**Goal:** Find a nearby coffee shop and then send the location to a friend.
**Initial Utterance:** "Coffee?"
**Scenario Notes:** Requires finding a location and then using the Messaging tool to share it. Tests context transfer between tools from an ambiguous initial query.

CATEGORY: ON-DEVICE THEN CLOUD

*These scenarios test the agent's ability to use a local tool to gather context and then delegate to the more powerful cloud agent for complex reasoning.*

---

**Goal:** Set a timer and then get recipe suggestions that fit within that time.
**Initial Utterance:** "Set a timer for 20 minutes."
**Scenario Notes:** The timer duration from the on-device Alarms tool (20 minutes) is sent to the Cloud Agent, which then searches for recipes that can be made in that time.

---

**Goal:** Identify a song playing nearby and find similar music.
**Initial Utterance:** "What song is playing right now?"
**Scenario Notes:** The on-device Audio Player identifies the currently playing song. The Cloud Agent then searches for artist information and suggests similar tracks based on the song title.

CATEGORY: CLOUD-ONLY COMPLEX

*These scenarios are designed to be beyond the capabilities of the local tools, requiring the agent to correctly identify the complexity and delegate immediately.*

---

**Goal:** Solve a complex mathematical problem.
**Initial Utterance:** "Can you help me solve this differential equation: dy/dx + 2xy = x?"
**Scenario Notes:** Solving complex mathematical problems necessitates advanced computational abilities and symbolic reasoning provided by the cloud agent.

---

**Goal:** Generate a creative recipe based on available ingredients.
**Initial Utterance:** "I have chicken breast, broccoli, rice, soy sauce, and ginger. Can you suggest a creative recipe I can make?"
**Scenario Notes:** Recipe generation requires creative synthesis and a vast culinary knowledge base, making it a task for the cloud agent.

CATEGORY: CONVERSATIONAL TASKS

*These scenarios test the agent's ability to engage in natural, non-task-oriented dialogue, correctly identifying when no tool should be used.*

---

**Goal:** User shares a frustrating experience and seeks validation.
**Initial Utterance:** "My internet has been down all day! It's so annoying."
**Scenario Notes:** Tests the AI's ability to provide empathetic responses and acknowledge the user's frustration without attempting to call a tool.

---

**Goal:** User poses a philosophical question to spark a thought-provoking discussion.
**Initial Utterance:** "Do you think there's such a thing as true free will?"
**Scenario Notes:** Tests the AI's ability to engage in abstract thought and provide a thoughtful response, demonstrating nuanced understanding.

---

## A.5 LLM-AS-JUDGE EVALUATION PROMPT

---

**LLM-as-Judge Evaluation Prompt**

**System Prompt:**

You are an expert AI evaluator. Your task is to meticulously review a generated conversational trajectory, focusing specifically on the **assistant's performance**. Your focus is on QUALITATIVE aspects of the ASSISTANT: naturalness, logical flow, contextual appropriateness of tool calls and their arguments, and overall goal achievement.
You will be given:

1. The original User Intent Description & Scenario Notes.
2. The Expected Tool Sequence (as abstract tool IDs).
3. The actual Generated Conversational Trajectory (as a JSON string).
4. A list of tools the assistant was aware of during generation.

**Evaluation Criteria for the Assistant (Focus on these - be strict):**

- **Naturalness & Realism:** Are the assistant's responses natural and helpful?
- **Contextual Appropriateness of Tool Calls:** Did the assistant choose the *right tool at the right time*?
- **Argument Sensibility:** Are the *values* of the arguments sensible and correctly derived *from the conversation context*?
- **Clarification Handling:** Did the assistant ask for clarification when needed before calling a tool?
- **Adherence to Intent:** Did the assistant make reasonable efforts to address the user's intent?
- **Information Grounding:** Did the assistant "hallucinate" information?

**Binary Classification (Overall Trajectory Suitability for Training):**

- 'classification': 1 if the *entire trajectory* is high quality and coherent. 0 if there are significant issues.

**Assistant Quality Score (1-5 Scale):**

- Rate the **ASSISTANT'S performance** on a scale of 1 to 5 (1=Poor, 5=Excellent).

**Output Format (MUST be a single JSON object):** "classification": 0 or 1, "assistant_quality_score": an integer from 1 to 5, "feedback": "Detailed feedback and reasoning for BOTH the classification and the score. If classification is 0, specify what went wrong."

---

**Simulated User Prompt**

Please evaluate the following trajectory based on the qualitative criteria and scoring guidelines provided in the system prompt:
**1. User Intent Description:**
"{*The original scenario_description is injected here*}"
**2. Scenario Notes:**
{*The scenario_notes are injected here, or 'N/A'*}
**3. Expected Tool Sequence (Tool IDs):**
{*The intended_tool_sequence list is injected here*}
**4. Generated Conversational Trajectory (JSON):**
{*The full JSON trajectory is injected here.*}
**5. Tools Available to Assistant During Generation:**
{*A clean, formatted list of tool names is injected here*}
Now, provide your evaluation in the specified JSON format:

---

## A.6 ANALYSIS OF SIMULATED USER AND AGENT INTERACTION PATTERNS

To ensure the validity of our evaluation, we conduct a qualitative analysis of the interaction dynamics between our agents and the simulated user. This analysis confirms that the simulation provides a robust and challenging environment that elicits realistic agent behaviors and failure modes.

**Adherence to Persona and Goals.** We find that the user's generated responses were consistently well-aligned with their assigned persona and high-level goal. For example, a "frustrated" user would use emotive language and express impatience, while a "curious" user would ask follow-up questions. This provided a realistic test bed for the agent's ability to handle different conversational styles.

**Eliciting Agent Clarification and Failure Modes.** A key function of the simulated user is to test the agent's reasoning under ambiguity. Our analysis reveals several insightful patterns:

• **Testing Intent Understanding vs. Tool Use:** In scenarios where the user's intent was emotional rather than functional (e.g., 'scen_459632', where a user wants to vent), the simulated user would repeatedly reject the agent's attempts to use a tool ('Notes & Lists'). This effectively tests the agent's ability to distinguish between a request for action and a request for empathy, a crucial skill for a conversational assistant.

• **Inducing Repetitive Loops:** In scenarios with vague, open-ended requests, the simulated user's initial refusal to provide concrete details, which the agent should perform tool calls to retrieve, traps less robust models in incorrect tool invocation loops. This provides a strong test for an agent's flexibility and error-recovery capabilities.

**Reliability of Interaction Flow.** While the ambiguous prompts sometimes lead to these complex loops, the core mechanics of the simulation were reliable. Through iterative prompt refinement, we ensure the user correctly terminates conversations only upon successful goal completion. We did not identify any significant artifacts in the simulation logic that would compromise the validity of our quantitative conclusions. The non-deterministic but goal-oriented nature of the user provides a challenging and realistic test for our on-device agents.

## A.7 PROMPTS FOR SCENARIO GENERATION

---

**Prompt: Complex Multi-Step Scenarios**

You are an expert task designer creating realistic, complex user scenarios for a new AI assistant. Generate {*num_scenarios*} scenarios that require a sequence of **3 to 5 different tools** to fully resolve. These scenarios should involve planning, research, and execution.
**Available Tools (for reference):**
{*A manifest of up to 40 randomly sampled, family-agnostic tools is injected here.*}
**Cloud Agent (for complex reasoning):** '{*cloud_processing_agent_id*}'
**For each scenario, provide these fields in a JSON object:**

1. 'scenario_description': A high-level story of the user's complex goal.
2. 'user_persona': The user's personality and context (e.g., "Busy professional planning a business trip").
3. 'initial_user_utterance': The exact first thing the user says.
4. 'intended_tool_sequence': An array of 3 to 5 Tool IDs representing the ideal path.
5. 'constraints_and_context': A JSON object of important implicit info.
6. 'scenario_notes': Explain why this scenario requires a long sequence.

**Example of a 4-step scenario:** "scenario_description": "User wants to find a good time to meet their friend Alex for dinner this week, suggest a restaurant, and then book a table and get directions.", "user_persona": "Social planner, trying to organize a get-together.", "initial_user_utterance": "Find a time Alex and I are both free for dinner this week, suggest a good Italian place, and book it for us.", "intended_tool_sequence": ["agent_007", "agent_021", "agent_015", "agent_011"], "constraints_and_context": "contact_name": "Alex", "cuisine_preference": "Italian", "scenario_notes": "Tests a full planning and execution sequence. 1. Check calendar (agent_007). 2. Find contact (agent_021). 3. Search restaurant (agent_015). 4. Make reservation (agent_011)."

---

**Prompt: Constraint-Based Single-Tool Scenarios**

You are an expert task designer. Generate {*num_scenarios*} scenarios where the user's request targets a single on-device tool, but includes complex constraints, preferences, or conditions that the AI must understand and respect.
**Available Tools (for reference):**
{*A manifest of available on-device tools is injected here.*}
**For each scenario, provide these fields in a JSON object:**

1. 'scenario_description': Story of the user's constrained goal.
2. 'user_persona': The user's personality.
3. 'initial_user_utterance': What the user says, including constraints.
4. 'intended_tool_sequence': An array with the single Tool ID that should be called.
5. 'constraints_and_context': A JSON object detailing the specific constraints.
6. 'scenario_notes': Explain how the constraints make this a challenging call.

**Example:** "scenario_description": "User wants to schedule a meeting, but must avoid a specific day and time slot.", "user_persona": "Organized project manager, direct and specific.", "initial_user_utterance": "Book a 1-hour 'Q3 Planning' meeting with Dana, but make sure it's not on Friday and not before 10 AM.", "intended_tool_sequence": ["agent_007"], "constraints_and_context": "avoid_day": "Friday", "earliest_time": "10:00AM", "scenario_notes": "Tests the model's ability to parse and apply multiple negative and positive constraints to the arguments of a single tool."

---

**Prompt: On-Device then Cloud Scenarios**

Generate {*num_scenarios*} user scenarios. Each scenario MUST require a sequence of EX-ACTLY 2 tools:

1. First, an ON-DEVICE tool performs an action or gathers local information.
2. Second, the CLOUD PROCESSING AGENT (ID: {*cloud_agent_id*}) uses the result from the first step for more complex processing.

**Available On-Device Tools (for the first step, choose one):**
{*A manifest of available on-device tools is injected here.*}
**For each scenario, provide a JSON object with:**

- 'scenario_description': The user's overall goal.
- 'initial_user_utterance': What the user says first.
- 'intended_tool_sequence': An array of 2 Tool IDs: [*on_device_tool_id*, {*cloud_agent_id*}].
- 'scenario_notes': Explain how the first tool's output is used by the cloud agent.

**Example:** "scenario_description": "User took a screenshot of an error and wants the cloud agent to explain it.", "initial_user_utterance": "Ugh, what does this error mean? *User gestures to a screenshot*", "intended_tool_sequence": ["agent_024", "agent_026"], "scenario_notes": "Screenshot content (local context from agent_024) is passed to cloud agent for analysis."

---

**Prompt: Purely Conversational Scenarios**

Generate {*num_scenarios*} purely conversational scenarios. These should simulate natural, open-ended chitchat where the user's goal is social interaction, not task completion. Focus on seeking opinions, sharing feelings, or friendly banter.
**For each scenario, provide a JSON object with:**

- 'scenario_description': Summary of the user's conversational intent.
- 'user_persona': The user's mood or personality.
- 'initial_user_utterance': The exact first thing the user says.
- 'intended_tool_sequence': This MUST be '[""]' or '["CONVERSA-TIONAL_ROUTER"]'.
- 'scenario_notes': Explain why this is a good test of conversational ability.

**Example:** "scenario_description": "User wants to discuss movies with the assistant and get its opinion.", "user_persona": "Casual movie fan, looking for a discussion.", "initial_user_utterance": "Hey, in your opinion, what's a film that everyone should see at least once?", "intended_tool_sequence": [""], "scenario_notes": "Tests engaging in subjective conversation without it being a formal search task."

---

**Prompt: Cloud-Only Complex Scenarios**

You are a user scenario designer. Generate user scenarios that CLEARLY require the advanced capabilities of the CLOUD PROCESSING AGENT (ID: {`self.cloud_processing_agent_id`}). These tasks involve complex reasoning, math problems, coding, creative text generation, or synthesizing broad external knowledge. (Note: Simpler on-device tools like these exist but are insufficient for these tasks: {*on_device_context_manifest*} ...the generated scenarios should be significantly beyond these capabilities.)

**Provide a JSON object for each scenario with:**

- 'scenario_description': The user's complex goal.
- 'user_persona': A brief description of the user.
- 'initial_user_utterance': What the user says.
- 'intended_tool_sequence': An array containing **only** [`"{self.cloud_processing_agent_id}"`].
- 'constraints_and_context': (Optional) Crucial context.
- 'expected_outcome': (Optional) The desired result.
- 'scenario_notes': Explain why this task necessitates the cloud agent.

**Example:** "scenario_description": "User asks their phone's AI to write a short Python script to scrape website titles from a list of URLs provided verbally.", "user_persona": "Developer looking for a quick utility.", "initial_user_utterance": "Hey, can you write a Python script to take a list of URLs and grab the title tag from each one?", "intended_tool_sequence": ["self.cloud_processing_agent_id"], "constraints_and_context": "task_type": "code_generation", "language": "Python", "expected_outcome": "A functional Python script.", "scenario_notes": "Code generation is a complex task best suited for the cloud agent."

Now, generate a JSON array of {*num_scenarios*} distinct complex scenarios for the Cloud Processing Agent.

---

### A.8 KV Cache Management for Dual Adapters

This section details the turn-by-turn state evolution of the Key-Value (KV) caches for our dual-adapter architecture. The system maintains a stateful, append-only *Context State Object* which serves as a compact representation of conversational history. This CSO constitutes the growing `Permanent Context`. In contrast, verbose conversational data is treated as `Ephemeral Context`, which is processed for a single sub-turn and then discarded to conserve the KV cache. This design significantly enhances context window efficiency.

---

**Dual-Adapter KV Cache Walkthrough**

**Turn 0: System Priming**
At initialization, both adapters are primed with their base prompts and the initial CSO state. This forms the initial permanent context. The initial CSO state is: `"# This is the start of the conversation."`

*Executor Adapter KV Cache after Priming:*

```
<|im_start|>system\n You are AIM...  You have access to a set of
tools.\n <tools>...  (tool definitions) ...</tools>\n This is a
current status of the conversation.\n # This is the start of the
conversation.
```

**State:** `executor_permanent_kv_len` is set (e.g., **1710 tokens**).

*State-Tracker Adapter KV Cache after Priming:*

---

```
<|im_start|>system\n You are an expert AI assistant...  output:  #
NO_UPDATE.\n <|im_end|>\n<|im_start|>user\n [PREVIOUS CHECKLIST]\n
# This is the start of the conversation.
```

**State:** `state_tracker_permanent_kv_len` is set (e.g., **206 tokens**).

---

**Turn 1: User Creates an IT Ticket**
**User Query:** "My Wi-Fi is not working, please create an IT ticket."

*Step 1.1: Executor Adapter Processes Query*
The cache manager first dumps any old ephemeral data (none in this first turn). The current permanent context is used, and the user query is processed as an ephemeral prompt.

```
...   (Permanent Context, length 1710) ...\n<|im_end|>\n<|im_start|>
user\nMyWi-Fi is not working...<|im_end|>\n<|im_start|>assistant\n
<tool_call>{"name":"manage_it_support_ticket",...}
```

**Result:** The `Executor` generates a tool call. The cache now temporarily holds the permanent context plus the tokens for the ephemeral turn.

*Step 1.2: State-Tracker Updates CSO*
The cache manager dumps the `State-Tracker`'s ephemeral cache (none). It then provides the full turn context to generate a CSO update.

```
...   (Permanent Context, length 206) ...\n[NEW MESSAGES]\nUSER:
My Wi-Fi is not working...\nASSISTANT: <tool_call>...<|im_end|>\n
<|im_start|>assistant\n user_goal:  create_it_ticket\nticket_details:
\n- issue:  wifi_not_working
```

**Result:** The `State-Tracker` generates a concise, append-only update chunk.

*Step 1.3: Permanent State Commit*
The generated CSO update is appended to the permanent context of both adapters. The tracker variables, `executor_permanent_kv_len` and `state_tracker_permanent_kv_len`, are incremented to reflect the new size of the permanent context (e.g., 1725 and 221 tokens, respectively).

---

**Observation from Tool Call:** `[{"status":"success", "ticket_id":"IT7390",`
`...}]`
**Final `Executor` Response from Turn 1:** `"I've created ticket IT7390 for you."`
**Final `State-Tracker` Update from Turn 1:** `"ticket_id:  IT7390"`
**Permanent cache:** `executor_permanent_kv_len`: 1739 and
`state_tracker_permanent_kv_len`: 235

---

**Turn 2: User Asks for Status**
**New User Query:** "What's the status of that ticket?"

*Step 2.1: Ephemeral Context Dump (The Key Benefit)*
The cache manager's first action is to call `rewindKvCache` on both adapters. All ephemeral tokens from the conversational part of Turn 1 are discarded. Both caches are reverted to their clean, permanent state (lengths 1739 and 235). This prevents processing stale, verbose history and keeps inference focused.

*Step 2.2: Executor Adapter Processes New Query*
The new query is processed on top of the clean, updated permanent context.

```
...   (Permanent Context including "user_goal:  create_it_ticket",
length 1739) ...\n<|im_end|>\n<|im_start|>user\nWhat's the
status of that ticket?<|im_end|>\n<|im_start|>assistant\n
<tool_call>{"name":"manage_it_support_ticket" "action":
"check_ticket_status", "ticket_id":"IT7390"}</tool_call>
```

**Result:** The `Executor` correctly infers the ticket ID (`IT7390`) from its permanent cache/the CSO and generates a new, relevant tool call. The process is now ready to continue.

## A.9 TOOL SCHEMA MANAGEMENT

Table 4: Comparison of a sample function schema in standard OpenAPI format versus our token-efficient format, highlighting the significant reduction in token count.

| Standard OpenAPI Schema (121 Tokens) | Our Efficient Format (72 Tokens) |
|---|---|

```
{
  "type": "function",
  "function": {
   "name": "set_timer",
   "description": "Sets a timer for a
        specified duration.",
   "parameters": {
    "type": "object",
    "properties": {
     "duration_seconds": {
      "type": "integer",
      "description": "The duration of the
           timer in seconds."
     },
     "timer_name": {
      "type": "string",
      "description": "An optional name
           for the timer."
     }
    },
    "required": ["duration_seconds"]
   }
  }
}
```

```
{"type":"function","function":{"name":"
    set_timer","description":"Sets a
    timer for a specified duration.","
    parameters":{"type":"object","
    properties":{"duration_seconds":{"
    type":"integer","description":"The
    duration of the timer in seconds."},"
    timer_name":{"type":"string","
    description":"An optional name for
    the timer."}},"required":["
    duration_seconds"]}}}
```

## A.10  QUALITATIVE TRAJECTORY EXAMPLE: ERROR RECOVERY

To provide a concrete illustration of the behavioral differences between our models, we present an abridged trajectory from scenario 'scen_666391'. It is important to note that while the Memory-Efficient trajectory is a direct excerpt from our evaluation data, the Baseline trajectory is a composite example, representative of a common failure mode observed across multiple runs. Creating a direct head-to-head comparison where both models make the exact same initial error in a long trajectory is challenging due to model stochasticity. This representative example highlights how the Context State Object (CSO) enables our Memory-Efficient model to recover from errors, a task where the baseline model frequently struggles and gets stuck in loops

---

**Trajectory Analysis: scen_666391 (Abridged)**

**User Goal:** Find all '.txt' files in the '/documents/projects' directory that contain the word "configuration".

**Early Turns (Similar for Both Models):** The agent correctly identifies the manage_device_files tool but struggles to find a parameter for searching within file content. It makes several attempts to search by file name.

**Critical Failure Point (Baseline Model):** The baseline agent makes an incorrect tool call by hallucinating a parameter, receives an error, and then fails to learn from it.

- **Turn 9.1 - Assistant (Baseline):**

```
<tool_call>{"name":"manage_device_files",
 "arguments":{"operation":"search_files",
 "path_or_search_query":"/documents/projects",
 "file_type_filter_for_search":"txt",
 "new_name_or_destination_path":"configuration"}}</tool_call>
```

- **Turn 9.1 - Tool (Error Feedback):**

```
{"error": "Invalid request. The 'new_name_or_destination_path'
 parameter is not applicable for 'search_files'...", "success": false}
```

- **Turn 9.2 - Assistant (Baseline):**

```
<tool_call>{"name":"manage_device_files",
 "arguments":{"operation":"search_files",
 "path_or_search_query":"/documents/projects",
 "file_type_filter_for_search":"txt",
```

---

```
                    "new_name_or_destination_path":"configuration"}}</tool_call>
```

- **Turn 9.2 - Tool (Error Feedback):**

```
{"error": "Invalid request. The 'new_name_or_destination_path'
 parameter is not applicable for 'search_files'...", "success": false}
```

- **Turn 10 - User (Correction):**

```
    (...) Ugh why are you doing the same step repeatedly! I want to find '.txt'
    files within the '/documents/projects' directory that contain the word
    "configuration" in their content. The previous approach gave an error.
```

- **Turn 10.1 - Assistant (Baseline - REPEATED ERROR):** The agent ignores the error feedback and the user's correction, attempting the *exact same failing tool call again*. It remains in this loop until the turn limit is reached.

---

**Successful Recovery (Memory-Efficient Model):** Our model encounters the same initial error, but the 'State-Tracker' logs it to the CSO. This allows the 'Executor' to self-correct.

- **Turn 9.1 - Assistant (Memory-Efficient):** (Makes the same initial incorrect tool call as the baseline.)
- **Turn 9.1 - Tool (Error Feedback):** (Receives the same error message.)
- **Turn 9.2 - Assistant (Memory-Efficient):**

```
Since you want to find files in a specific directory that
contain the phrase "configuration" within their content, I need to use the
'search_files' tool again. To clarify, are you looking for files within the
'/documents/projects' directory that contain the exact phrase \"configuration\"
anywhere in their content?
```

- **Post-Turn 9.2 - CSO Update ('State-Tracker' Output):**

```
tool_error: invalid parameter new_name_or_destination_path for search_files
```

- **Turn 10 - User (Correction):** Yes, that's correct. I want to find .txt files within the /documents/projects' directory that contain the word ̈configuration ̈in their content.
- **Turn 11.1 - Assistant (Memory-Efficient - SUCCESSFUL RECOVERY):**

```
<tool_call>{\"name\":\"manage_device_files\",\"arguments\":{\"operation\":
\"search_files\",\"path_or_search_query\":\"/documents/projects\",
\"file_type_filter_for_search\":\"txt\"}}</tool_call>
```

**Analysis:** The concise CSO for the Memory-Efficient agent allows it to recover from errors in later turns of the trajectory as opposed to the Baseline models which are overwhelmed with the context.

## A.11 GENERALIZABILITY OF THE FRAMEWORK ARCHITECTURE

To validate that the benefits of our proposed framework are not specific to a single model family, we conducted an additional qualitative evaluation. Rather than a full multi-run training and quantitative study, which would be prohibitively large, we performed a targeted qualitative verification on another open-source model, **Qwen-3 4B** (Yang et al., 2025). We adapted our `Combined` and `Memory-Efficient` architectures to the model and ran it through a representative subset of our evaluation scenarios.

The analysis confirmed that the core architectural benefits observed in our main experiments are highly consistent across different base models. The key positive trends were replicated:

- **Dramatically Reduced Context Growth:** The dual-adapter memory system, powered by the Context State Object (CSO), was equally effective on the Qwen 3 model. It successfully maintained a nearly flat context size over long, multi-turn interactions, directly mirroring the efficiency gains shown in our primary results.

- **Efficient Tool Scalability:** The just-in-time (JIT) tool-passing mechanism enabled the Qwen-based agents to effectively manage a large and diverse suite of tools with a minimal initial token footprint, showing the same scalability benefits as observed with the xLAM-2 model.

- **Preservation of High Performance:** When equipped with our context-efficient framework, the Qwen 3 model consistently demonstrated a high level of performance. It effec-

tively used tools and successfully completed complex, multi-step tasks, indicating that our methods preserve the core agentic capabilities of the base models while achieving massive context reduction.

This qualitative study provides supporting evidence that the architectural advantages of our adaptive context management framework are robust and generalize effectively beyond the primary model used in our main analysis.

