# OpenReview forum: "Efficient On-Device Agents via Adaptive Context Management"
_ICLR.cc/2026/Conference — ICLR 2026 Conference Withdrawn Submission_

### Official Review · Reviewer_CQ3w · 2025-10-31

**Soundness:** 2
**Presentation:** 3
**Contribution:** 3
**Rating:** 4
**Confidence:** 2

**Summary:**

This paper proposes a novel framework for on-device agents, focusing on the scalability issue regarding tool usage. Within the framework, the agent handles the requests from the users in two stages: orchestration and context management, which is backed up by a specially fine-tuned LoRA adapter. Benefiting from the design and fine-tuning, the agents exhibit effectiveness in the evaluation tasks and demonstrate desired traits, such as reduced history log, often functioning as a bottleneck in the long term. The authors also present a detailed analysis of the architectural designs.

**Strengths:**

The strengths of this paper are as follows:
1. Framework design: The proposed framework is well-thought-out. Each choice of the framework is well supported with reasonable observations/insights. Especially, the choices are highly technically detailed at the low level, while the appropriateness remains clear at the high level.
2. Balanced metric: The authors combine various metrics to examine the efficacy of the proposed method. The combination of precision (i.e., rule-based) and LLM-as-judge (i.e., model-based) provides a synergy for examining the agents more rigorously.
3. Analysis: The experiments are studied extensively. The authors provide detailed explanations of the results with rationalizations of such results in the view of agents’ behaviors. The detailed study on the architectural trade-off is also noticeable.

**Weaknesses:**

The weaknesses/questions/suggestions of this paper are as follows:
1. Reference: There are missing parts of the Appendix that are referenced in the main text (e.g., Appendix A.5 & A.6).
2. Tasks:
	- In the line 827-828 (Appendix A.2.6): does this mean that prior benchmarks are ill-posed, or the framework is designed in a wrong manner? The “our models will not work” part sounds very unconvincing. Would there be any method to adapt the existing framework (e.g., changing the context part with some special module with a subset of problems of the existing benchmark)?
	- While the authors claim that simulated users present reality, the claim is not supported enough. I request the authors to elaborate on the simulated
users.
	- (Minor) Why do the authors restrict the use of tools in conversational tasks? My interpretation is that the agents might use the tool for more natural conversation (e.g., exploring notes to understand users better).
3. Baselines: I believe that more baselines using other methods are in demand. (If more baselines are not applicable, please justify why.) The presented baselines are mostly ablation studies, rather than comparisons with other methods, hindering judgment on the superiority of the proposed methods over other approaches. Especially, because there is a large gap between xLAM-2 3B and Baseline, but marginal differences between from Baselines to Combined, judging the effectiveness of the proposed method is highly difficult.

Overall, I believe that the paper has a strong framework design, while the results are relatively weak to support its superiority. I believe that this can also be resolved by adding more tasks, which is less appealing compared to adding more baselines.

**Questions:**

(See above)

---

### Official Review · Reviewer_FUyE · 2025-10-31

**Soundness:** 3
**Presentation:** 3
**Contribution:** 2
**Rating:** 6
**Confidence:** 3

**Summary:**

This paper addresses the challenge of limited memory in device-based systems with context-efficient on-device agents. It not only reduces context overhead but also maintains performance on complex tasks. The framework introduced dual LoRA adapters tos compress conversational history into a structured, append-only logs thus significantly reduces context growth. Further improvements minimizes initial prompt size by using a minimalist schema format and a just-in-time mechanism to load full tool definitions only when needed.

**Strengths:**

- Novel Context Management: The CSO system effectively balances memory efficiency and task fidelity by leveraging structured logging and semantic compression. This addresses the critical bottleneck of long-context degradation in on-device settings.
- Practical Tool Optimization: The minimalist schema format and JIT mechanism drastically reduce initial token overhead, enabling agents to handle more tools within constrained memory budgets.
- Strong Empirical Results: The experiments show significant improvements in context efficiency without sacrificing performance on complex tasks like multi-tool orchestration and cloud delegation.

**Weaknesses:**

- The experiment is conducted on a 3B-parameter xLAM 2. While a brief test on Qwen-3 4B suggests scalability, the results may not extend to other architectures or larger models without retraining.
- The JIT schema-passing mechanism introduces 500ms latency per turn for the CSO update cycle on a Galaxy S25 CPU, which would  increase latency in time-sensitive applications.
- While the paper provides detailed methodology, it is still difficult for researchers to reproduce it and it would be better for authors to add an additional code repository.

**Questions:**

Please see Weaknesses.

---

### Official Review · Reviewer_fuCP · 2025-10-31

**Soundness:** 3
**Presentation:** 3
**Contribution:** 2
**Rating:** 4
**Confidence:** 3

**Summary:**

This paper studies the context compression for on-device AI agents by proposing a framework with three components: (1) a dual-adapter memory system where one LoRA adapter handles tasks while another compresses conversation history into an append-only CSO in key-value format, (2) a token-efficient tool schema format, and (3) a JIT tool loading mechanism that initially presents only tool names/descriptions and loads full schemas only after selection. Experiment results show the effectiveness of the proposed method.

**Strengths:**

1. Context constraints on resource-limited devices are a genuine deployment bottleneck overlooked by long-context research.
2. The paper is well written and easy to follow.
3. Empirical results show the proposed method achieves 10-25× context reduction with maintained or improved performance.

**Weaknesses:**

1. **Limited Novelty in Context Compression**: Context compression is a mature field with extensive prior work. The proposed Context State Object is essentially task-specific summarization in key-value format, not a fundamentally new compression paradigm. The dual-adapter architecture uses standard techniques. The paper needs to better clarify what makes this approach fundamentally different from applying existing summarization methods to agent conversations, as the primary contribution appears to be system-level engineering rather than methodological innovation.
2. **Insufficient Baseline Comparisons**: The paper lacks comparisons with established context management methods. Without these baselines, it's unclear whether efficiency gains stem from CSO's specific design or simply from having any compression mechanism. The custom evaluation protocol further limits comparability with prior work.
3. **Unvalidated LLM-as-Judge**: The qualitative evaluation relies on Gemini 2.0 Flash without reliability validation. The paper provides no human agreement analysis, calibration study, or bias assessment. Given significant score variations across models, it remains unclear whether these differences reflect actual quality gaps or judgment artifacts.

**Questions:**

See weaknesses.

---

### Official Review · Reviewer_dzNG · 2025-11-01

**Soundness:** 2
**Presentation:** 2
**Contribution:** 2
**Rating:** 2
**Confidence:** 4

**Summary:**

This paper present an engineering solution for managing the context of on-device agents, which is a critical challenge given the limited memory to hold the context on device. The proposed method mainly focus on summarizing tool execution context and tracing the dialog state. Results verify the methods’ effectiveness and efficiency.

**Strengths:**

1. This paper studies a practical scenarios and can be helpful for the deployment of llm.
2. The proposed method is simple and easy to understanding while being effective.

**Weaknesses:**

1. The technical contribution might be minor. The author lacks a comprehensive discussion on existing method and their limitations. Therefore, It is hard to justify the novelty of the proposed method.
2. The author only compares their method to few baselines on a self-curated datasets, which is a weak evaluation.
3. Although the author study on-device memory management, no specific memory parameters and computation resources available for a number of representative devices are provided. Therefore, it is hard to understand whether the proposed method satisfy the practical needs.

**Questions:**

See above

---

### Note · Authors · 2025-11-21

**Comment:**

We thank the reviewers for their feedback. To present the contribution at its full potential, we plan to expand the analysis and comparative evaluation. We are therefore withdrawing the submission and will prepare an improved version.

**Withdrawal Confirmation:**

I have read and agree with the venue's withdrawal policy on behalf of myself and my co-authors.